# Dynamical analysis and optimal control of the developed information transmission model

**Sida Kang**[1], **Xilin Hou**[1]*, **Yuhan Hu**[2], **Hongyu Liu**[3]

**1** Department of Electronic and Information Engineering, University of Science and Technology Liaoning, Anshan, Liaoning, China, **2** Department of Science, University of Science and Technology Liaoning, Anshan, Liaoning, China, **3** Department of Business Administration, University of Science and Technology Liaoning, Anshan, Liaoning, China

* hou_xilinDFS@163.com

**Data Availability Statement:** All relevant data are within the paper.

**Funding:** This article is supported by National Natural Science Foundation of China (NNSFC) grant 71472080, Humanities and Social Sciences

## Abstract

Information transmission significantly impacts social stability and technological advancement. This paper compares the phenomenon of "Super transmission" and "Asymptomatic infection" in COVID-19 transmission to information transmission. The former is similar to authoritative information transmission individuals, whereas the latter is similar to individuals with low acceptance in information transmission. It then constructs an $S2EIR$ model with transmitter authority and individual acceptance levels. Then, it analyzes the asymptotic stability of information-free and information-existence equilibrium on a local and global scale, as well as the model's basic reproduction number, $R_0$. Distinguished with traditional studies, the population density function and Hamiltonian function are constructed by taking proportion of "Super transmitter" and proportion of hesitant group turning into transmitters as optimization control variables. Based on the Pontryagin maximum principle, an optimal control strategy is designed to effectively facilitate information transmission. The numerical simulation corroborates the theoretical analysis results and the system's sensitivity to control parameter changes. The research results indicate that the authoritative "Super transmitter" has a beneficial effect on information transmission. In contrast, the "Asymptomatic infected individual" with poor individual acceptance level negatively affects information transmission.

## 1 Introduction

Information is a necessary component of human society's development and has significant impacts on human life. In terms of the impact of information on human society, it can be broadly classified as either positive or negative. For instance, knowledge transmission [1, 2] and innovation capability transmission [3] are beneficial to social development, which constitutes positive information, whereas the spread of rumors [4, 5] and computer viruses [6–8] constitutes negative information. It can be classified into two broad categories of information transmission channels: contact transmission [9, 10] and network transmission [11, 12]. As a result, it is necessary to investigate the mechanisms of information transmission and control.

Research Projects of Education Department of Liaoning Province China grant 2020LNJC11 and LJKZ0311.

**Competing interests:** The authors have declared that no competing interests exist.

The information transmission characteristics are strikingly similar to those of infectious disease and rumor transmission [13, 14]. Thus, the information transmission models are frequently enhanced using infectious disease and rumor transmission models. The spread of infectious diseases was the first spreading problem studied by scholars. Infectious disease models that are frequently used include the *SI* model, the *SIS* model, and the *SIR* model [15–17]. Daley and Kendal developed a model of rumor transmission based on the classical infectious disease model. Scholars refer to this model as the DK model [18]. Based on the classical model of rumor transmission, scholars proposed the *SEIR* model with lurkers [19], the *SIVR* model with mutants [20], the *SIVRS* model with restorers [21], the *SIHR* model with forgetting and memory mechanisms [22], the *SHIR* model with hesitation mechanisms [23], and the *SLIS* model with age-structured [24].

In the last five years, scholars have conducted in-depth studies on information transmission from the following perspectives: (1) in terms of information type. Liu et al. developed a new *SEIR* model for heterogeneous networks to better understand the dynamics of information transmission in microblogs [25]. Wan et al. proposed the *SIB* model for analyzing the transmission of information about e-commerce discounts on a scale-free network [26]. Hosseini et al. developed the *SEIRS-QV* model with vaccination and isolation strategies to investigate malware transmission behavior in heterogeneous networks while also accounting for additional influencing factors such as user perception and network delay [27]. He et al. hypothesized that there is competition for various types of information in online social networks. They proposed the *CISIR* model to shed light on information competition and transmission [28]. Xiao et al. considered the dynamic changes in anti-rumor information from different aspects. They developed an evolutionary game theory-based driving mechanism for information, and ultimately proposed the *SKIR* model of rumor and anti-rumor competition [29]. (2) in terms of individual transmission, Zhang et al. examined the coexistence of rumors and authoritative information in social networks. They proposed the $IS_1 S_2 C_1 C_2 R_1 R_2$ model, which includes a super transmitter, a super authoritative information transmitter, a rumor suppressor, and a super authoritative information suppressor [30]. Li et al. believed that education had a significant impact on information transmission in a multi-language and heterogeneous network environment. As a result of this issue, the $I_k S_k^1 S_k^2 R_k^1 R_k^2$ model was proposed [12]. Sang et al. hypothesized that social network users were heterogeneous and that individual perception behaviors influenced information transmission. As a result of this phenomenon, the *SEIRD* model was proposed that takes into account individual consciousness, social relationships, and knowledge levels [31]. Additionally, Fu et al. investigated the transmission of e-commerce discount information via social networks, believing that the super transmitter had a significant promotional effect on information and could increase potential users' degree of acceptance of information. As a result, they proposed the *SEIAR* model, which takes into account super transmitters and potential users [32]. Yin et al. conducted an in-depth analysis of opinion leaders' influence on the information transmission process using Weibo data and proposed an *OD-SFI* model [33]. Zhang et al. developed the *SETQR* model by considering time delay, trust, and forgetting mechanisms [34]. Additionally, the transmission of COVID-19's information and the virus has been a hot topic for scholars in recent years [35, 36]. Abdo et al. analyzed and found the solution for the model of nonlinear fractional differential equations describing the deadly and most parlous virus, so-called coronavirus (COVID-19). The study discovered that the susceptibility decreases more rapidly at the lower fractional order of the derivative. Similarly, the increase in infections is also rapid, but in a smaller order [37]. Almalahi et al. used fractal-ABC type fractional differential equations by incorporating population self-protection behavior changes to study the dynamics of 2019-nCoV transmission [38], and investigated

sufficient conditions of existence and uniqueness of positive solutions for a finite system of $\psi$-Hilfer fractional differential equations [39]. Jeelani et al. investigated a fractional-order mathematical model of COVID-19 [40]. These research findings are critical in the prediction of 2019-nCoV.

We have a certain understanding of the characteristics and modes of information transmission after analyzing the information transmission mechanism. Some scholars have conducted additional research on information transmission control to convert information into a controllable variable. Wang et al. developed a model of alcoholism's transmission dynamics. They used prevention effectiveness, treatment costs, media coverage, and contact ratio as control variables to achieve optimal control of the alcoholism problem [41]. Alzahrani et al. also used a similar approach to research on smoking prevention strategies [42]. Huo discovered that psychological factors have a significant influence on information transmission and that scientific knowledge and official information can be used to guide people's psychological activities. As a result, the two factors are used as control variables to mitigate the spread of false information [43]. It can generally use network control strategies such as forced silence, information labeling, administrator control, and controlling the relative density of individual nodes to suppress online information transmission [4, 44–46].

The scholars mentioned above have conducted extensive research on transmitting various types of information via various channels, including the transmission of rumors [4], false information [5], competitive information [28] and malware [27]. Scholars who consider information transmission from the perspective of individuals are usually based on the attitude of transmission individuals to an event [22–24, 31, 32]. Few scholars consider the transmission individuals' attributes. Generally, most scholars primarily focus on controlling and transmitting negative information in networks [44, 46]. On the other hand, confidential information must be transmitted via the contact for information that must be taught in person, such as high technology or practical skills. At the same time, due to the unique characteristics of this type of information, as well as the requirements for the transmitter's authority and the recipient's ability to accept it, there is a shortage of literature on the subject. In addition, the spread of COVID-19 is a hot topic in current research. However, few scholars consider the phenomenon of "Asymptomatic infection" in the transmission model. The research on the transmission of the phenomenon of "Asymptomatic infection" is even less, especially in information transmission. Therefore, consider the phenomenon of "Asymptomatic infection" in the information transmission model and investigate the impact of "Asymptomatic infection" on the information transmission system.

As a result of the above considerations, this paper compares the phenomenon of "Super transmission" and "Asymptomatic infection" in COVID-19 transmission to information transmission. It then proposes an *S2EIR* model that incorporates the phenomena of "Super transmission" and "Asymptomatic infection". During the transmission of COVID-19, the phenomena of "Super transmission" and "Asymptomatic infection" are common. The term "Super transmission" refers to the phenomenon in which highly contagious individuals are more likely to spread the virus to the majority of patients. The term "Asymptomatic infection" refers to a situation in which a patient has been infected with a virus and has become a carrier of the virus but does not exhibit obvious disease symptoms due to individual resistance and physical quality differences. Both of these phenomena frequently occur during information transmission. "Super transmitters" are analogous to authoritative individuals in information transmission. The information transmission by such individuals is more easily accepted. Whereas "Asymptomatic infected individuals" are analogous to hesitant individuals with a low level of acceptance. However, this group of people has certain infectivity in the transmission of COVID-19. However, individuals who conceal information rarely choose to share it with

others during the transmission of information. At the same time, others are unaware that such individuals know the information. This is also the manifestation of "Asymptomatic" in information transmission. The optimal control strategy of information transmission is quantified by scientific methods to test these phenomena.

The remaining sections of this paper are organized as follows. In Section 2, an $S2EIR$ model is constructed that considers the phenomena of "Super transmission" and "Asymptomatic infection". In Section 3, it will propose the basic reproduction number, $R_0$, and demonstrate the locally and globally asymptotically stability of information-free equilibrium and information-existence equilibrium. Section 4 proposes the optimal control existence and control strategy of information transmission. Section 5, through numerical simulation, analyzes the influence of parameter changes on information transmission, as well as the effect of the optimal control strategy. Section 6 analyzes sensitivity analysis on important control parameter changes in information transmission. The last section provides the conclusion.

## 2 The model

This paper discusses the concept of an open virtual community. The population size is variable at any point in time $t$, and the total population size can be represented by $N(t)$. Each group can be classified into one of five categories. (1) The easy group, which is not exposed to information but is receptive to it, is represented by $S(t)$. (2) $E_1$ represents the hesitant group that has a high level of individual acceptance as a result of the influence of the "Super transmitter" $E_1(t)$. (3) The hesitant group is represented by $E_2(t)$, which exhibits the characteristics of "Asymptomatic infection" in the absence of "Super transmitter" influence. (4) $I(t)$ represents the group that has a high level of individual acceptance and transmits information. (5) $R(t)$ denotes the group without adoption that has a high level of individual acceptance due to external factors' influence, or that has a low level of individual acceptance as a result of their own factors.

In the constructed model, the authoritative transmitter has a strong ability to interpret information, increasing the degree to which receivers adopt information. However, some individuals transmit information without the assistance of a "Super transmitter". Due to the difference in an individual's comprehension ability or learning level, information transmitters also have poor information interpretation. As a result, such individuals are unable to develop an effective understanding of information. Hence, these individuals are unable to disseminate information widely.

This is slightly different from the virus transmission phenomenon referred to as "Asymptomatic infection". Although the "Asymptomatic infected individuals" of information transmission have received information, they are unable to use it effectively, let alone transmit it to other individuals, due to their insufficient understanding and acceptance of information.

In order to reflect the phenomena in information transmission. An $S2EIR$ model is constructed that considers the phenomena of "Super transmission" and "Asymptomatic infection" in this paper. The model flow diagram is given in Fig 1.

The parameters in the $S2EIR$ model can be explained as follows:

- The number of individuals in a social system generally varies over time. As a result, $B$ is defined in this paper as the number of immigrants in the social system. Simultaneously, some individuals in the social system may withdraw due to various force majeure factors; thus, $\mu$ is used in this paper to define the removal rate;

- When information begins to transmit in the system, the easy adopters will make contact with the transmitter with a certain probability, and the contact rate is defined as $\alpha$. There is a subgroup of information transmitters known as "Super transmitters" that has a proportion

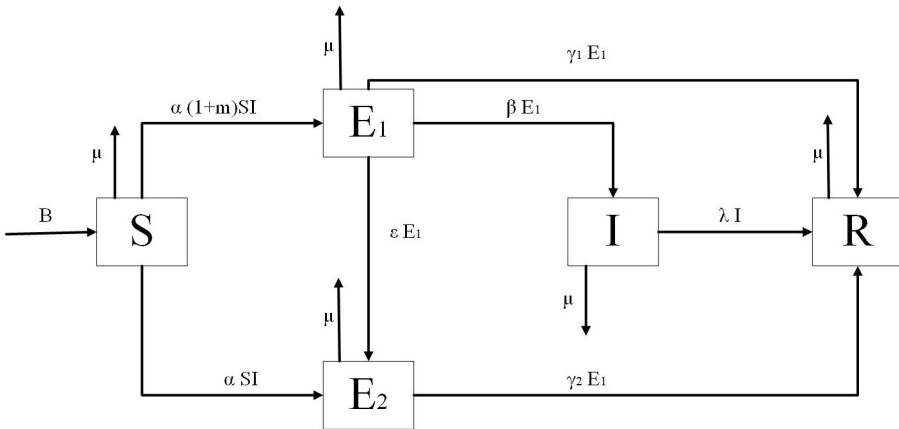

**Fig 1. The flow diagram of the model.**

of $m$. Simultaneously, the "Super transmitters" will increase the degree of transmission, converting the easy adopters into a hesitant group of strong recipients with a probability of $\alpha(1 + m)$;

- When the hesitant group of strong recipients adopts the information through their own understanding and learning, this group will be transformed into the transmitters with a probability of $\beta$. On the contrary, if their learning ability is insufficient, they will become the hesitant people of "Asymptomatic infection" with the probability of $\varepsilon$;

- Due to the difference in individual acceptance degrees, the two hesitant groups will become the non-acceptance groups with the probability of $\gamma_1$ and $\gamma_2$, respectively. Meanwhile, because the periodicity and timeliness of information constrain the transmitter, it will not transmit such information. Thus, the recovery rate is defined as $\lambda$.

The parameters of $S2EIR$ model are summarized in Table 1.

**Table 1. The parameter description of S2EIR model.**

| Parameter | Description |
| --- | --- |
| $S(t)$ | The number of easy adopters at the time $t$. |
| $E_1(t)$ | The number of hesitating individuals that has a high level of individual acceptance as a result of the influence of the "Super transmitter" at the time $t$. |
| $E_2(t)$ | The number of hesitating individuals that exhibits the characteristic of "Asymptomatic infection" in the absence of "Super transmitter" influence at the time $t$. |
| $I(t)$ | The number of transmittable individuals at the time $t$. |
| $R(t)$ | The number of not adopters at the time $t$. |
| $\alpha$ | Information transmitting rate. |
| $m$ | The proportion of "Super transmitters". |
| $\beta$ | Transition probability from state $E_1$ to state $I$. |
| $\varepsilon$ | Transition probability from state $E_1$ to state $E_2$. |
| $\gamma_1$ | Transition probability from state $E_1$ to state $R$. |
| $\gamma_2$ | Transition probability from state $E_2$ to state $I$. |
| $\mu$ | Removal rate per unit time. |
| $B$ | The number of immigrants in the social system per unit time. |

Based on the above analysis, we construct an *S2EIR* model that considers the phenomena of "Super transmission" and "Asymptomatic infection". The system dynamics equations are described as follows:

$$
\begin{cases}
\dfrac{dS}{dt} = B - \alpha(1+m)SI - \alpha SI - \mu S, \\[2mm]
\dfrac{dE_1}{dt} = \alpha(1+m)SI - \beta E_1 - \gamma_1 E_1 - \varepsilon E_1 - \mu E_1, \\[2mm]
\dfrac{dE_2}{dt} = \alpha SI + \varepsilon E_1 - \gamma_2 E_2 - \mu E_2, \\[2mm]
\dfrac{dI}{dt} = \beta E_1 - \lambda I - \mu I, \\[2mm]
\dfrac{dR}{dt} = \lambda I + \gamma_1 E_1 + \gamma_2 E_2 - \mu R.
\end{cases}
\tag{1}
$$

Where:

$$
B > 0, \mu > 0, \alpha > 0, \beta > 0, \gamma_1 > 0, \gamma_2 > 0, \varepsilon > 0, \lambda > 0, m \in (0, 1],
\tag{2}
$$

and

$$
S(t) + E_1(t) + E_2(t) + I(t) + R(t) = N(t).
\tag{3}
$$

## 3 Stability analysis of the model

Firstly, the basic reproduction number $R_0$ of system (1) approach in the next generation matrix [47]. In this paper, $R_0$ represents the number of next generation from a single information transmitter produced.

Let $X = (E_1, I, R, E_2, S)^T$, then system (1) can be written as:

$$
\frac{dX}{dt} = F(X) - V(X),
\tag{4}
$$

$$
F(X) = \begin{pmatrix} \alpha(1+m)SI \\ \beta E_1 \\ 0 \\ 0 \\ 0 \end{pmatrix}, \quad
V(X) = \begin{pmatrix} \beta E_1 + \gamma_1 E_1 + \varepsilon E_1 + \mu E_1 \\ \lambda I + \mu I \\ -\lambda I - \gamma_1 E_1 - \gamma_2 E_2 + \mu R \\ -\alpha SI - \varepsilon E_1 + \gamma_2 E_2 + \mu E_2 \\ -B + \alpha(1+m)SI + \alpha SI + \mu S \end{pmatrix}.
\tag{5}
$$

Calculate the Jacobian matrices of $F(X)$ and $V(X)$ in system (5) respectively, and then take the sub matrices corresponding to the first two variables (i.e. E1, I) directly related to the number of communicators. The results are as follows:

$$
F = \begin{pmatrix} 0 & \alpha(1+m)S \\ \beta & 0 \end{pmatrix}, \quad
V = \begin{pmatrix} \beta + \gamma_1 + \varepsilon + \mu & 0 \\ 0 & \lambda + \mu \end{pmatrix},
\tag{6}
$$

where $F$ and $V$ represent the infection and transition matrices respectively [38]. By simple

calculation, the inverse matrix of V can be obtained as:

$$V^{-1} = \begin{pmatrix} \dfrac{1}{\beta + \gamma_1 + \varepsilon + \mu} & 0 \\ 0 & \dfrac{1}{\lambda + \mu} \end{pmatrix}. \tag{7}$$

The next generation matrix [47] is

$$FV^{-1} = \begin{pmatrix} 0 & \dfrac{\alpha(1+m)S}{\lambda + \mu} \\ \dfrac{\beta}{\beta + \gamma_1 + \varepsilon + \mu} & 0 \end{pmatrix}. \tag{8}$$

Hence, the basic reproduction number $R_0$ of system (1) is the spectral radius of the next generation matrix $FV^{-1}$. Here, the spectral radius is the maximum value of characteristic root of $FV^{-1}$. Therefore, $R_0$ can be computed as:

$$R_0 = \rho(FV^{-1}) = \sqrt{\frac{B\alpha\beta(1+m)}{\mu(\beta + \gamma_1 + \varepsilon + \mu)(\lambda + \mu)}}. \tag{9}$$

Then, the information will disappear if $R_0 < 1$, and the information-free equilibrium point of system (1) can be easily observed as $E^0 = \left( {}^B\!/\!_\mu, 0, 0, 0, 0 \right)$.

While the information will be transmitted if $R_0 > 1$, and the information-existence equilibrium point of system (1) can be obtained as $E^* = (S^*, E_1^*, E_2^*, I^*, R^*)$. $E^*$ must satisfy the following equations:

$$\begin{cases} B - \alpha(1+m)S^*I^* - \alpha S^*I^* - \mu S^* = 0, \\[2mm] \alpha(1+m)S^*I^* - \beta E_1^* - \gamma_1 E_1^* - \varepsilon E_1^* - \mu E_1^* = 0, \\[2mm] \alpha S^*I^* + \varepsilon E_1^* - \gamma_2 E_2^* - \mu E_2^* = 0, \\[2mm] \beta E_1^* - \lambda I^* - \mu I^* = 0, \\[2mm] \lambda I^* + \gamma_1 E_1^* + \gamma_2 E_2^* - \mu R^* = 0. \end{cases} \tag{10}$$

Solving the equations of Eq (10), we can get the information-existence equilibrium point $E^* = (S^*, E_1^*, E_2^*, I^*, R^*)$:

$$S^* = \frac{B}{\mu R_0^2},\tag{11}$$

$$E_1^* = \frac{B(1+m)(R_0^2 - 1)}{(2+m)(\beta + \gamma_1 + \varepsilon + \mu)R_0^2},\tag{12}$$

$$E_2^* = \frac{B(\beta + \gamma_1 + \varepsilon + \mu)(R_0^2 - 1) + B\varepsilon(1+m)(R_0^2 - 1)}{(2+m)(\gamma_2 + \mu)(\beta + \gamma_1 + \varepsilon + \mu)R_0^2},\tag{13}$$

$$I^* = \frac{\mu(R_0^2 - 1)}{\alpha(2+m)}.\tag{14}$$

**Theorem 3.1** *If $R_0 < 1$, the information-free equilibrium point $E^0 = \left({}^B/_\mu, 0, 0, 0, 0\right)$ of system (1) is locally asymptotically stable.*

*Proof of Theorem 3.1* The Jacobin matrix of system (1) at information-free equilibrium point $E^0 = \left({}^B/_\mu, 0, 0, 0, 0\right)$ can be written as:

$$J(E^0) = \begin{bmatrix} -\mu & 0 & 0 & -\frac{B}{\mu}\alpha(2+m) & 0 \\ 0 & -(\beta + \gamma_1 + \varepsilon + \mu) & 0 & \frac{B}{\mu}\alpha(1+m) & 0 \\ 0 & \varepsilon & -\gamma_2 + \mu & \frac{B}{\mu}\alpha & 0 \\ 0 & \beta & 0 & -(\lambda + \mu) & 0 \\ 0 & \gamma_1 & \gamma_2 & \lambda & -\mu \end{bmatrix}.\tag{15}$$

It is easy to know that $J(E^0)$ has three negative eigenvalues $\Lambda_{01} = \Lambda_{02} = -\mu < 0$, $\Lambda_{03} = -(\gamma_2 + \mu) < 0$, and the other eigenvalues are the characteristic roots of $|hE\text{-}J(E^0)|$, where:

$$|hE - J(E^0)| = \begin{vmatrix} h + (\beta + \gamma_1 + \varepsilon + \mu) & -\frac{B}{\mu}\alpha(1+m) \\ -\beta & h + (\lambda + \mu) \end{vmatrix}.\tag{16}$$

The eigenvalues of Eq (16) can be obviously obtained as:

$$\begin{aligned} \Lambda_{04} &= \frac{-[(\beta + \gamma_1 + \varepsilon + \mu) + (\lambda + \mu)]}{2} \\ &+ \frac{\sqrt{[(\beta + \gamma_1 + \varepsilon + \mu) + (\lambda + \mu)]^2 - 4(1 - R_0^2)(\beta + \gamma_1 + \varepsilon + \mu)(\lambda + \mu)}}{2}, \end{aligned}\tag{17}$$

and

$$
\begin{aligned}
\Lambda_{05} \quad = \quad & \frac{-[(\beta + \gamma_1 + \varepsilon + \mu) + (\lambda + \mu)]}{2} \\
& - \frac{\sqrt{[(\beta + \gamma_1 + \varepsilon + \mu) + (\lambda + \mu)]^2 - 4(1 - R_0^2)(\beta + \gamma_1 + \varepsilon + \mu)(\lambda + \mu)}}{2}.
\end{aligned}
\tag{18}
$$

If $R_0 < 1$, so $\Lambda_{04} < 0$ and $\Lambda_{05} < 0$. Hence, the information-free equilibrium point $E^0 = \left( {}^B/_\mu, 0, 0, 0, 0 \right)$ of system (1) is locally asymptotically stable if $R_0 < 1$ based on the Routh-Hurwitz criterion.

**Theorem 3.2** *If $B\alpha(2 + m) \leq \mu^2$, the information-free equilibrium point $E^0 = \left( {}^B/_\mu, 0, 0, 0, 0 \right)$ of system (1) is globally asymptotically stable.*

*Proof of Theorem 3.2* It is easy to know that $S(t) + E_1(t) + E_2(t) + I(t) + R(t) = N(t)$ and satisfy $\frac{dN(t)}{dt} \leq B - \mu S(t)$. It illustrates that:

$$
\limsup_{t \to 0} N(t) \leq \frac{B}{\mu}.
\tag{19}
$$

For $t \geq 0$, the positive invariant set of system (1) can be written as:

$$
T = \left\{ (S(t), E_1(t), E_2(t), I(t), R(t)) \in R_5^+ : S(t) + E_1(t) + E_2(t) + I(t) + R(t) \leq \frac{B}{\mu} \right\}.
\tag{20}
$$

Then, we construct the Lyapunov function $L(t) = E_1(t) + E_2(t) + I(t) + R(t)$ and $L'(t)$ can be computed as:

$$
\begin{aligned}
L'(t) \quad = \quad & \alpha(1 + m)SI - \beta E_1 - \gamma_1 E_1 - \varepsilon E_1 - \mu E_1 + \alpha SI + \varepsilon E_1 \\
& - \gamma_2 E_2 - \mu E_2 + \beta E_1 - \lambda I - \mu I + \lambda I + \gamma_1 E_1 + \gamma_2 E_2 - \mu R \\
= \quad & [\alpha(2 + m)S - \mu]I - \mu(E_1 + E_2 + R) \leq \left[ \alpha(2 + m)\frac{B}{\mu} - \mu \right]I \\
& - \mu(E_1 + E_2 + R),
\end{aligned}
\tag{21}
$$

obviously, $L'(t) \leq 0$ if $S \leq \frac{B}{\mu}$ and $B\alpha(2 + m) \leq \mu^2$.

In addition, $L'(t) = 0$ holds if and only if $S(t) = S^0$, $E_1 = E_2 = I = R = 0$. From system (1), it is known that $E^0$ is the only solution in $T$ when $L'(t) = 0$. Therefore, based on the Lyapunov-LaSalle Invariance Principle [48], it is shown that every solution of system (1) approach $E^0$ for $t \to \infty$. Hence, the information-free equilibrium point $E^0 = \left( {}^B/_\mu, 0, 0, 0, 0 \right)$ of system (1) is globally asymptotically stable.

**Theorem 3.3** *If $R_0 > 1$, the information-existence equilibrium point $E^* = (S^*, E_1^*, E_2^*, I^*, R^*)$ of system (1) is locally asymptotically stable.*

*Proof of Theorem 3.3* The Jacobin matrix of system (1) at information-existence equilibrium point $E^* = (S^*, E_1^*, E_2^*, I^*, R^*)$ can be written as:

$$J(E^*) = \begin{bmatrix} -\alpha(1+m)I^* - \alpha I^* - \mu & 0 & 0 & -\alpha(1+m)S^* - \alpha S^* & 0 \\ \alpha(1+m)I^* & 0 & 0 & \alpha(1+m)S^* & 0 \\ \alpha I^* & -(\beta + \gamma_1 + \varepsilon + \mu) & -(\gamma_2 + \mu) & \alpha S^* & 0 \\ 0 & \varepsilon & 0 & -(\lambda + \mu) & 0 \\ 0 & \gamma_1 & \gamma_2 & \lambda & -\mu \end{bmatrix}. \quad (22)$$

It is easy to know that two of the negative eigenvalues of system (22) are $\Lambda_{11} = -\mu$ and $\Lambda_{12} = -(\gamma_2 + \mu)$, and the other eigenvalues are the characteristic roots of $|hE\text{-}J(E^*)|$, where:

$$|hE - J(E^*)| = \begin{vmatrix} h + \alpha(1+m)I^* + \alpha I^* + \mu & 0 & \alpha(1+m)S^* + \alpha S^* \\ -\alpha(1+m)I^* & h + (\beta + \gamma_1 + \varepsilon + \mu) & -\alpha(1+m)S^* \\ 0 & -\beta & h + (\lambda + \mu) \end{vmatrix}. \quad (23)$$

We can obviously obtain the eigenvalues of Eq (23), where:

$$\begin{aligned} |hE - J(E^*)| &= h^3 + [(\beta + \gamma_1 + \varepsilon + \mu) + (\lambda + \mu) + \alpha(2 + m)I^* + \mu]h^2 \\ &\quad + [(\beta + \gamma_1 + \varepsilon + \mu)(\lambda + \mu) + \alpha(2 + m)(\beta + \gamma_1 + \varepsilon + \mu) \\ &\quad + \lambda + \mu)I^* + (\beta + \gamma_1 + \varepsilon + \mu + \lambda + \mu)\mu - \alpha\beta(1 + m)S^*]h \\ &\quad + [\alpha(2 + m)(\beta + \gamma_1 + \varepsilon + \mu)(\lambda + \mu)I^* + (\beta + \gamma_1 + \varepsilon \\ &\quad + \mu)(\lambda + \mu)\mu - \alpha\beta\mu(1 + m)S^*]. \end{aligned} \quad (24)$$

Then we construct a cubic polynomial and replace the coefficient with $a_3, a_2, a_1, a_0$ to determine the other eigenvalues of system (22). Hence, Eq (24) can be rewritten as:

$$a_3 h^3 + a_2 h^2 + a_1 h + a_0 = 0, \quad (25)$$

where:

$$a_3 = 1, \quad (26)$$

$$a_2 = (\beta + \gamma_1 + \varepsilon + \mu) + (\lambda + \mu) + \mu R_0^2, \quad (27)$$

$$a_1 = (\beta + \gamma_1 + \varepsilon + \mu)(\lambda + \mu) + (\beta + \gamma_1 + \varepsilon + \mu + \lambda + \mu)\mu R_0^2 - \frac{B\alpha\beta(1 + m)}{\mu R_0^2}, \quad (28)$$

$$a_0 = (\beta + \gamma_1 + \varepsilon + \mu)(\lambda + \mu)\mu R_0^2 - \frac{B\alpha\beta(1 + m)}{R_0^2}, \quad (29)$$

and

$$
\begin{aligned}
a_2 a_1 - a_3 a_0 &= (\beta + \gamma_1 + \varepsilon + \mu)(\lambda + \mu)(\beta + \gamma_1 + \varepsilon + \mu + \lambda + \mu) + (\beta + \gamma_1 \\
&+ \varepsilon + \mu + \lambda + \mu)^2 \mu R_0^2 + (\beta + \gamma_1 + \varepsilon + \mu + \lambda + \mu)\mu^2 R_0^4 \\
&+ \frac{B\alpha\beta(1+m)}{R_0^2} - \frac{B\alpha\beta(1+m)(\beta + \gamma_1 + \varepsilon + \mu + \lambda + \mu)}{\mu R_0^2} - B\alpha\beta(1+m).
\end{aligned}
\tag{30}
$$

The condition of information-existence equilibrium point $E^* = (S^*, E_1^*, E_2^*, I^*, R^*)$ is locally asymptotically stable and the conditions: (i) $a_3, a_2, a_1, a_0 > 0$ and (ii) $a_2 a_1 - a_3 a_0 > 0$ based on the Routh-Hurwitz criterion. It is easy to know that $a_3, a_2 > 0$.

If $\mu^2 R_0^4 > \alpha\beta(1+m)$ and $R_0 > 1$, then $a_1, a_0 > 0$ and $a_2 a_1 - a_3 a_0 > 0$. In this case, the Routh-Hurwitz criterion are satisfied. Hence, the information-existence equilibrium point $E^* = (S^*, E_1^*, E_2^*, I^*, R^*)$ of system (1) is locally asymptotically stable.

**Theorem 3.4** *If $R_0 > 1$, the information-existence equilibrium point $E^* = (S^*, E_1^*, E_2^*, I^*, R^*)$ of system (1) is globally asymptotically stable.*

*Proof of Theorem 3.4* We construct the Lyapunov function as:

$$
W(t) = [(S(t) - S^*) + (E_1(t) - E_1^*) + (E_2(t) - E_2^*) + (I(t) - I^*) + (R(t) - R^*)]^2,
\tag{31}
$$

and

$$
\begin{aligned}
W'(t) &= 2[(S(t) - S^*) + (E_1(t) - E_1^*) + (E_2(t) - E_2^*) + (I(t) - I^*) \\
&+ (R(t) - R^*)][S'(t) + E_1'(t) + E_2'(t) + I'(t) + R'(t)] \\
&= 2[(S(t) - S^*) + (E_1(t) - E_1^*) + (E_2(t) - E_2^*) + (I(t) - I^*) \\
&+ (R(t) - R^*)][B - \mu S - \mu E_1 - \mu E_2 - \mu I - \mu R].
\end{aligned}
\tag{32}
$$

Because of the existence of $E^* = (S^*, E_1^*, E_2^*, I^*, R^*)$, we can know that $B - \mu S^* - \mu E_1^* - \mu E_2^* - \mu I^* - \mu R^* = 0$, i.e., $B = \mu S^* + \mu E_1^* + \mu E_2^* + \mu I^* + \mu R^*$.

Then, Eq (32) can be computed as:

$$
\begin{aligned}
W'(t) &= 2[(S(t) - S^*) + (E_1(t) - E_1^*) + (E_2(t) - E_2^*) + (I(t) - I^*) + R(t) \\
&- R^*)][\mu S^* + \mu E_1^* + \mu E_2^* + \mu I^* + \mu R^* - \mu S - \mu E_1 - \mu E_2 - \mu I - \mu R] \\
&= -2[(S - S^*) + (E_1 - E_1^*) + (E_2 - E_2^*) + (I - I^*) + (R - R^*)]^2 \le 0.
\end{aligned}
\tag{33}
$$

Besides that, $W'(t) = 0$ holds if and only if $S(t) = S^*, E_1(t) = E_1^*, E_2(t) = E_2^*, I(t) = I^*, R(t) = R^*$. Hence, the information-existence equilibrium point $E^* = (S^*, E_1^*, E_2^*, I^*, R^*)$ of system (1) is globally asymptotically stable based on Lyapunov-LaSalle Invariance Principle [48].

## 4 The optimal control model

Two control objectives are proposed to expand the scope of information transmission based on the information transmission model established above. On the one hand, the number of information transmitters grows, and the information transmitted to the maximum. On the other hand, it has an increasing number of hesitant groups capable of comprehending information in depth and choosing to transmit it to expand the group of information transmitters.

Hence, the two proportion constants $m$ and $\beta$ in the model are transformed into control variables $m(t)$ and $\beta(t)$. The control variable $m(t)$ is used to control the proportion of authoritative "Super transmitters" in the crowd. More individuals in these groups will participate in information transmission through approaches such as policy guidance or macro-control. Thus, groups exposed to information are more receptive to it and can easily become new transmitters. The control variable $\beta(t)$ is used to control the proportion of hesitant groups turning into transmitters. The acceptance degree of these groups can be improved using media publicity or strengthening education. At the same time, they can be influenced by "Super transmitters" and become information transmitters more easily. In this model, the control variables $0 \leq m(t) \leq 1$ and $0 \leq \beta(t) \leq 1$. While $m(t) = 1$ and $\beta(t) = 1$, it means that the control effect is optimal and the information can be transmitted to the greatest extent. On the contrary, while $m(t) = 0$ and $\beta(t) = 0$, it means that the control measures are completely ineffective.

Hence, we propose an objective function as:

$$J(m, \beta) = \int_0^{t_f} \left[ E_1(t) + I(t) - \frac{c_1}{2} m^2(t) - \frac{c_2}{2} \beta^2(t) \right] dt, \tag{34}$$

and satisfy the state system as:

$$\begin{cases} \dfrac{dS}{dt} = B - \alpha(1 + m(t))SI - \alpha SI - \mu S, \\[2mm] \dfrac{dE_1}{dt} = \alpha(1 + m(t))SI - \beta(t)E_1 - \gamma_1 E_1 - \varepsilon E_1 - \mu E_1, \\[2mm] \dfrac{dE_2}{dt} = \alpha SI + \varepsilon E_1 - \gamma_2 E_2 - \mu E_2, \\[2mm] \dfrac{dI}{dt} = \beta(t)E_1 - \lambda I - \mu I, \\[2mm] \dfrac{dR}{dt} = \lambda I + \gamma_1 E_1 + \gamma_2 E_2 - \mu R. \end{cases} \tag{35}$$

The initial conditions for system (35) are satisfied:

$$S(0) = S_0, E_1(0) = E_{1,0}, E_2(0) = E_{2,0}, I(0) = I_0, R(0) = R_0, \tag{36}$$

where:

$$m(t), \beta(t) \in U \stackrel{\Delta}{=} \{(m, \beta) | (m(t), \beta(t)) \, measurable, 0 \leq m(t), \beta(t) \leq 1, \forall t \in [0, t_f]\}, \tag{37}$$

while $U$ is the admissible control set. The time interval of control is between 0 and $t_f$. $c_1$ and $c_2$ are positive weight coefficients shown the control strength and importance of two control measures.

**Theorem 4.1** *An optimal control pair $(m^*, \beta^*) \in U$ exists so that the function is established below*:

$$J(m^*, \beta^*) = \max\{J(m, \beta) : (m, \beta) \in U\}. \tag{38}$$

*Proof of Theorem 4.1* Let $X(t) = (S(t), E_1(t), E_2(t), I(t), R(t))^T$ and

$$L(t; X(t), m(t), \beta(t)) = E_1(t) + I(t) - \frac{c_1}{2} m^2(t) - \frac{c_2}{2} \beta^2(t). \tag{39}$$

The existence of an optimal pair must satisfy: (i) the set of control variables and state variables is nonempty, (ii) the control set $U$ is convex and closed, (iii) the right-hand side of the state system is bounded by a linear function in the state and control variables, (iv) the integrand of the objective functional is convex on $U$, (v) there exist constants $d_1, d_2 > 0$ and $\rho > 1$ such that the integrand of the objective functional satisfies:

$$- L(t; X(t), m; \beta) \geq d_1(|m|^2 + |\beta|^2)^{\rho/2} - d_2. \tag{40}$$

Conditions (i)-(iii) is clearly established, we just prove the condition (iv) and (v). One can easily obtain inequality:

$$S' \leq B, E_1{'} \leq \alpha(1+m(t))SI, E_2{'} \leq \alpha SI + \varepsilon E_1, I' \leq \beta(t)E_1, R' \leq \lambda I + \gamma_1 E_1 + \gamma_2 E_2. \tag{41}$$

Hence, condition (iv) is established. Then, for any $t \geq 0$, there is a positive constant $M$ which is satisfied $|X(t)| \leq M$, therefore

$$-L(t; X(t), m; \beta) = \frac{c_1 m^2(t) + c_2 \beta^2(t)}{2} - E_1(t) - I(t) \geq d_1(|m|^2 + |\beta|^2)^{\rho/2} - 2M. \tag{42}$$

Let $d_1 = \min\{\frac{c_1}{2}, \frac{c_2}{2}\}$, $d_2 = 2M$ and $\rho = 2$, then condition (v) is established. Hence, the optimal control can be realized.

**Theorem 4.2** *For the optimal control pair $(m^*, \beta^*)$ of state system* (35), *there exist adjoint variables $\delta_1, \delta_2, \delta_3, \delta_4, \delta_5$ that satisfy:*

$$\begin{cases} \dfrac{d\delta_1}{dt} = (\delta_1 - \delta_2)\alpha(1 + m(t))I + (\delta_1 - \delta_3)\alpha I + \delta_1 \mu, \\[2mm] \dfrac{d\delta_2}{dt} = 1 + (\delta_2 - \delta_4)\beta(t) + (\delta_2 - \delta_5)\gamma_1 + (\delta_2 - \delta_3)\varepsilon + \delta_2 \mu, \\[2mm] \dfrac{d\delta_3}{dt} = (\delta_3 - \delta_5)\gamma_2 + \delta_3 \mu, \\[2mm] \dfrac{d\delta_4}{dt} = 1 + (\delta_1 - \delta_2)\alpha(1 + m(t))S + (\delta_1 - \delta_3)\alpha S + (\delta_4 - \delta_5)\lambda + \delta_4 \mu, \\[2mm] \dfrac{d\delta_5}{dt} = \delta_5 \mu. \end{cases} \tag{43}$$

*With boundary conditions:*

$$\delta_1(t_f) = \delta_2(t_f) = \delta_3(t_f) = \delta_4(t_f) = \delta_5(t_f) = 0. \tag{44}$$

*In addition, the optimal control pair $(m^*, \beta^*)$ of state system* (35) *can be given by:*

$$m^*(t) = \min\left\{1, \max\left\{0, \frac{(\delta_1 - \delta_2)\alpha SI}{c_1}\right\}\right\}, \tag{45}$$

$$\beta^*(t) = \min\left\{1, \max\left\{0, \frac{(\delta_2 - \delta_4)E_1}{c_2}\right\}\right\}. \tag{46}$$

*Proof of Theorem 4.2* Define a Hamiltonian function enlarged with penalty term to obtain the expression of optimal control system and optimal control pair. The Hamiltonian function

enlarged can be written as:

$$
\begin{aligned}
H =\;& -E_1(t) - I(t) + \frac{c_1}{2} m^2(t) + \frac{c_2}{2} \beta^2(t) + \delta_1 [B - \alpha(1 + m(t))SI - \alpha SI - \mu S] \\
&+ \delta_2 [\alpha(1+m(t))SI - \beta(t)E_1 - \gamma_1 E_1 - \varepsilon E_1 - \mu E_1] + \delta_3 [\alpha SI + \varepsilon E_1 - \gamma_2 E_2 - \mu E_2] \\
&+ \delta_4 [\beta(t)E_1 - \lambda I - \mu I] + \delta_5 [\lambda I + \gamma_1 E_1 + \gamma_2 E_2 - \mu R] - \omega_{11} m(t) - \omega_{12}(1 - m(t)) \\
&- \omega_{21} \beta(t) - \omega_{22}(1 - \beta(t)),
\end{aligned}
\tag{47}
$$

which the penalty term is $\omega_{ij}(t) \geq 0$, and it is satisfied that $\omega_{11}(t)m(t) = \omega_{12}(t)(1 - m(t)) = 0$ at optimal control $m^*$ and $\omega_{21}(t)\beta(t) = \omega_{22}(t)(1 - \beta(t)) = 0$ at optimal control $\beta^*$.

Based on the Pontyragin maximum principle, the adjoint system can be written as:

$$
\frac{d\delta_1}{dt} = -\frac{\partial H}{\partial S}, \frac{d\delta_2}{dt} = -\frac{\partial H}{\partial E_1}, \frac{d\delta_3}{dt} = -\frac{\partial H}{\partial E_2}, \frac{d\delta_4}{dt} = -\frac{\partial H}{\partial I}, \frac{d\delta_5}{dt} = -\frac{\partial H}{\partial R},
\tag{48}
$$

and the boundary conditions of adjoint system are

$$
\delta_1(t_f) = \delta_2(t_f) = \delta_3(t_f) = \delta_4(t_f) = \delta_5(t_f) = 0.
\tag{49}
$$

Let $m^*$ as an example to give the optimality conditions. One have

$$
\frac{\partial H}{\partial m} = c_1 m(t) - \delta_1 \alpha SI + \delta_2 \alpha SI - \omega_{11} + \omega_{12} = 0,
\tag{50}
$$

and the optimal control formulae can be written as:

$$
m^* = \frac{(\delta_1 - \delta_2)\alpha SI}{c_1} + \omega_{11} - \omega_{12}.
\tag{51}
$$

To obtain the final optimal control formulae without $\omega_{11}$ and $\omega_{12}$ need to consider the following three situations.

The first situation is that $\omega_{11}(t) = \omega_{12}(t) = 0$ in set $\{t | 0 < m^*(t) < 1\}$, then the optimal control formulae can be written as:

$$
m^*(t) = \frac{1}{c_1}(\delta_1 - \delta_2)\alpha SI.
\tag{52}
$$

The second situation is that $\omega_{11}(t) = 0$ in set $\{t | m^*(t) = 1\}$, then the optimal control formulae can be written as:

$$
1 = m^*(t) = \frac{1}{c_1}[(\delta_1 - \delta_2)\alpha SI - \omega_{12}].
\tag{53}
$$

Due to $\omega_{12}(t) \geq 0$, it is shown that $\frac{1}{c_1}(\delta_1 - \delta_2)\alpha SI \geq 1$.

The third situation is that $\omega_{12}(t) = 0$ in set $\{t | m^*(t) = 0\}$, then the optimal control formulae can be written as:

$$
0 = m^*(t) = \frac{1}{c_1}[(\delta_1 - \delta_2)\alpha SI + \omega_{11}].
\tag{54}
$$

Based on the above situation, the final optimal control formulae of $m^*(t)$ can be written as $m^*(t) = \min\{1, \max\{0, \frac{(\delta_1 - \delta_2)\alpha SI}{c_1}\}\}$. Similarly, the final optimal control formulae of $\beta^*(t)$ can be written as $\beta^*(t) = \min\{1, \max\{0, \frac{(\delta_2 - \delta_4)E_1}{c_2}\}\}$.

So far, we get the optimal control system includes state system (35) with the initial conditions $S(0)$, $E_1(0)$, $E_2(0)$, $I(0)$, $R(0)$ and the adjoint system (43) with boundary conditions with the optimization conditions. The optimal control system can be written as:

$$
\begin{cases}
\dfrac{dS}{dt} = B - \alpha\left(1 + \min\left\{1, \max\left\{0, \dfrac{(\delta_1 - \delta_2)\alpha SI}{c_1}\right\}\right\}(t)\right)SI - \alpha SI - \mu S, \\[2ex]
\dfrac{dE_1}{dt} = \alpha\left(1 + \min\left\{1, \max\left\{0, \dfrac{(\delta_1 - \delta_2)\alpha SI}{c_1}\right\}\right\}(t)\right)SI \\[2ex]
\qquad - \min\left\{1, \max\left\{0, \dfrac{(\delta_2 - \delta_4)E_1}{c_2}\right\}\right\}(t)E_1 - \gamma_1 E_1 - \varepsilon E_1 - \mu E_1, \\[2ex]
\dfrac{dE_2}{dt} = \alpha SI + \varepsilon E_1 - \gamma_2 E_2 - \mu E_2, \\[2ex]
\dfrac{dI}{dt} = \min\left\{1, \max\left\{0, \dfrac{(\delta_2 - \delta_4)E_1}{c_2}\right\}\right\}(t)E_1 - \lambda I - \mu I, \\[2ex]
\dfrac{dR}{dt} = \lambda I + \gamma_1 E_1 + \gamma_2 E_2 - \mu R, \\[2ex]
\dfrac{d\delta_1}{dt} = (\delta_1 - \delta_2)\alpha\left(1 + \min\left\{1, \max\left\{0, \dfrac{(\delta_1 - \delta_2)\alpha SI}{c_1}\right\}\right\}(t)\right)I + (\delta_1 - \delta_3)\alpha I + \delta_1\mu, \\[2ex]
\dfrac{d\delta_2}{dt} = 1 + (\delta_2 - \delta_4)\min\left\{1, \max\left\{0, \dfrac{(\delta_2 - \delta_4)E_1}{c_2}\right\}\right\}(t) \\[2ex]
\qquad + (\delta_2 - \delta_5)\gamma_1 + (\delta_2 - \delta_3)\varepsilon + \delta_2\mu, \\[2ex]
\dfrac{d\delta_3}{dt} = (\delta_3 - \delta_5)\gamma_2 + \delta_3\mu, \\[2ex]
\dfrac{d\delta_4}{dt} = 1 + (\delta_1 - \delta_2)\alpha\left(1 + \min\left\{1, \max\left\{0, \dfrac{(\delta_1 - \delta_2)\alpha SI}{c_1}\right\}\right\}(t)\right)S \\[2ex]
\qquad + (\delta_1 - \delta_3)\alpha S + (\delta_4 - \delta_5)\lambda + \delta_4\mu, \\[2ex]
\dfrac{d\delta_5}{dt} = \delta_5\mu,
\end{cases}
\tag{55}
$$

and

$$
S(0) = S_0, E_1(0) = E_{1,0}, E_2(0) = E_{2,0}, I(0) = I_0, R(0) = R_0, \tag{56}
$$

$$
\delta_1(t_f) = \delta_2(t_f) = \delta_3(t_f) = \delta_4(t_f) = \delta_5(t_f) = 0. \tag{57}
$$

## 5 Numerical simulations

In this section, some numerical simulations will be given by the Rung-Kutta algorithm. The results of numerical simulation show that the rationality of the theoretical. Due to the range of

the parameters has not been explicitly given in previous studies. Therefore, this paper combining with the values of basic regeneration number $R_0$ and stability conditions, and give the numerical values of the parameters in the model.

In order to verify the locally and globally asymptotically stability of information-free equilibrium in Theorem 3.1 and Theorem 3.2. Let $B = 1$, $\alpha = 0.2$, $m = 0.3$, $\mu = 0.3$, $\gamma_1 = 0.5$, $\gamma_2 = 0.7$, $\beta = 0.6$, $\varepsilon = 0.2$, $\lambda = 0.4$. Through calculation, it can be concluded that $R_0 = 0.681 < 1$. Fig 2 verifies the stability of the model and shows that variety groups eventually converge to 0 change over time.

In order to verify the locally and globally asymptotically stability of information-existence equilibrium in Theorem 3.3 and Theorem 3.4. Let $B = 1$, $\alpha = 0.5$, $m = 0.3$, $\mu = 0.3$, $\gamma_1 = 0.5$, $\gamma_2 = 0.7$, $\beta = 0.6$, $\varepsilon = 0.2$, $\lambda = 0.2$. Through calculation, it can be concluded that $R_0 = 1.275 > 1$. Fig 3 verifies the stability of the model and shows that variety groups eventually converge to $E^*$ change over time.

In order to reveal the influence of optimal control pair $(m^*, \beta^*)$ on variety groups when we adopt the optimal control strategy. We give the image of "optimal control $(m = m^*(t), \beta = \beta^*(t))$", "middle control measure", "single control measure" and "constant control measure" respectively.

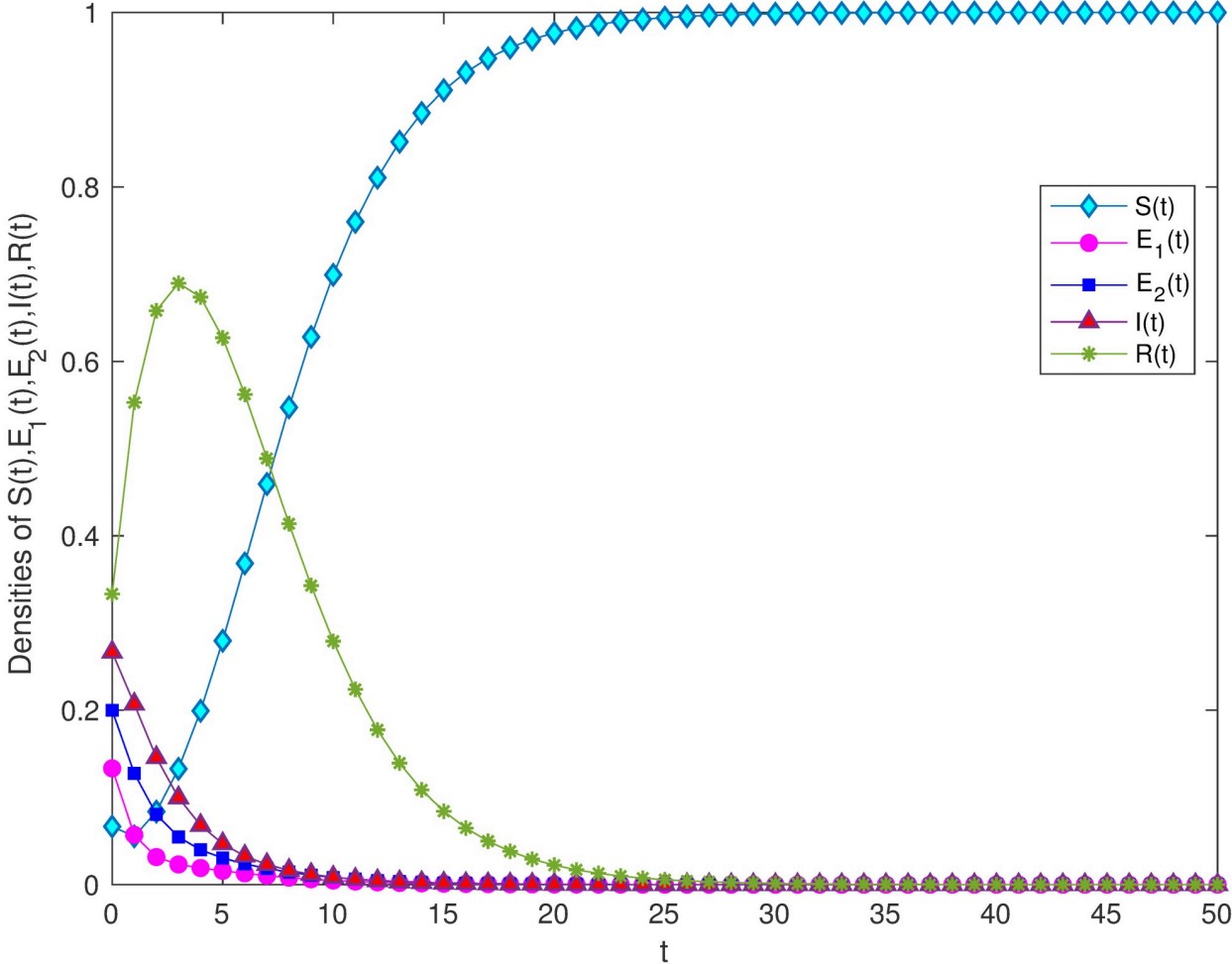

**Fig 2. The stability of information-free equilibrium $E^0$ of system 1 with $R_0 < 1$.**

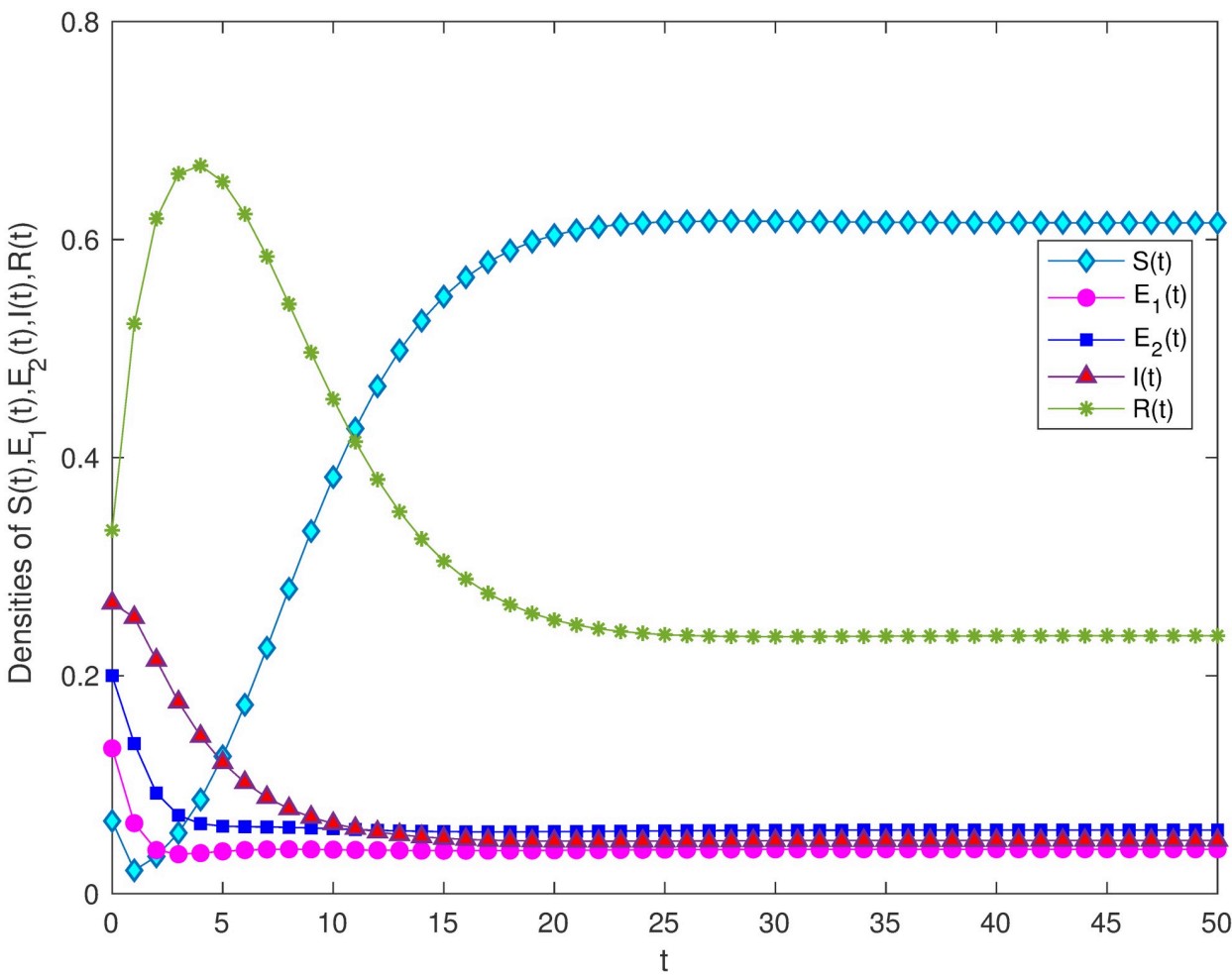

**Fig 3. The stability of information-existence equilibrium $E^*$ of system 1 with $R_0 > 1$.**

Firstly, let $B = 1$, $\alpha = 0.5$, $\mu = 0.3$, $\gamma_1 = 0.5$, $\gamma_2 = 0.7$, $\varepsilon = 0.2$, $\lambda = 0.2$ and different control strategies are adopted at the same time. Fig 4A–4C illustrate the densities of $E_1(t)$, $E_2(t)$, $I(t)$ change over time under different control strategies. The densities of $E_1(t)$, $E_2(t)$, $I(t)$ show that "optimal control" is better than "middle control measure" batter than "single control measure" batter than "constant control measure" in Fig 4. It illustrates that the information is effectively extended when the optimal control strategy is adopted. However, the density of $E_2(t)$ should be negative correlation with information extension in the theoretical analysis. It is inconsistent with the phenomenon shown in Fig 4B. Therefore, the further analysis is needed.

Then, let $B = 5$, $\alpha = 0.8$, $\mu = 0.2$, $\gamma_1 = 0.5$, $\gamma_2 = 0.7$, $\varepsilon = 0.2$, $\lambda = 0.2$ and different control strategies are adopted at the same time. Fig 5A–5C illustrate the densities of $E_1(t)$, $E_2(t)$, $I(t)$ change over time under different control strategies. The density of $E_1(t)$ can achieve the optimal state when "middle control measure" is adopted shown in Fig 5A. The density of $E_2(t)$ is less than the "middle control measure" and "single control measure $\beta$" when optimal control strategy is adopted shown in Fig 5B. It illustrates that the density of $E_2(t)$ is gradually showed a negative correlation with information extension with the increase of $B$. And Fig 5C illustrates the density of I(t) can achieve the optimal state when optimal control strategy is adopted.

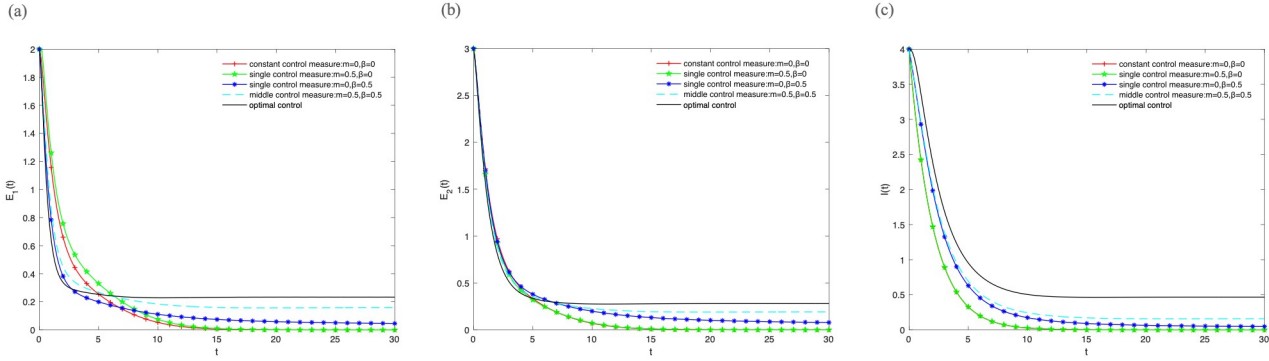

**Fig 4. The densities of (A) $E_1(t)$, (B) $E_2(t)$, (C) $I(t)$ change over time under different control strategies, where $B = 1$, $\alpha = 0.5$, $\mu = 0.3$, $\gamma_1 = 0.5$, $\gamma_2 = 0.7$, $\varepsilon = 0.2$, $\lambda = 0.2$.**

But we further find that the densities of $E_1(t)$, $E_2(t)$, $I(t)$ are approaching 0 when $\beta = 0$. Therefore, let $\beta$ increases to 0.06 when the values of other parameters remain unchanged. Then, the densities of $E_2(t)$, $I(t)$ change over time shown in Fig 6. Finally, the density of $E_2(t)$ reaches the minimum and the density of $I(t)$ reaches the maximum when the optimal control strategy is adopted. The information has been effectively transmitted and the density of $E_2(t)$ acted the minimum hindrance.

Based on the above analysis, the density of $I(t)$ can always achieve the optimal state when taking the optimal control strategy in any case. Meanwhile, the influence of the density of $E_2(t)$ on information transmission decreases gradually when taking the optimal control strategy with the increase of $B$. These phenomena demonstrate that the information has been effectively transmitted when the optimal control strategy is adopted for optimal control pair $(m^*, \beta^*)$.

## 6 Sensitivity analysis

To discuss the effect of control variables $m$ and $\beta$ on the basic reproductive number $R_0$, we need to perform the sensitivity analysis of $R_0$. According to the deduction above, we can figure

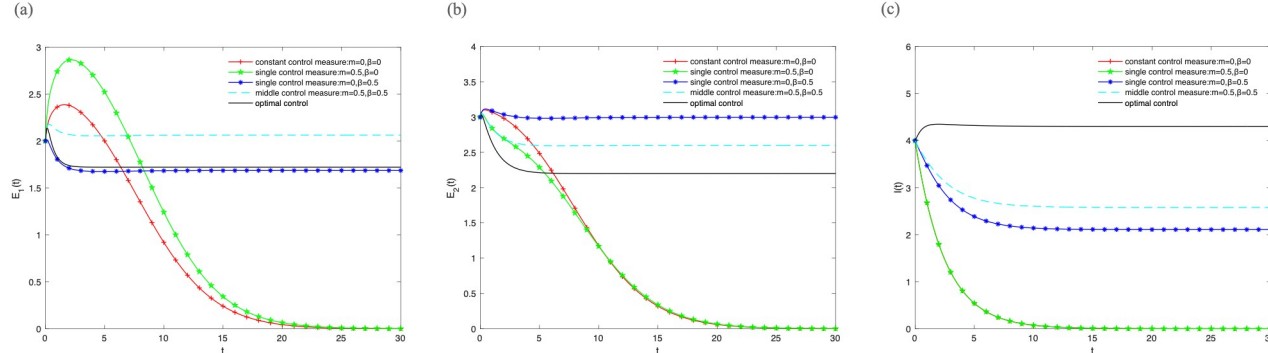

**Fig 5. The densities of (A) $E_1(t)$, (B) $E_2(t)$, (C) $I(t)$ change over time under different control strategies, where $B = 5$, $\alpha = 0.8$, $\mu = 0.2$, $\gamma_1 = 0.5$, $\gamma_2 = 0.7$, $\varepsilon = 0.2$, $\lambda = 0.2$.**

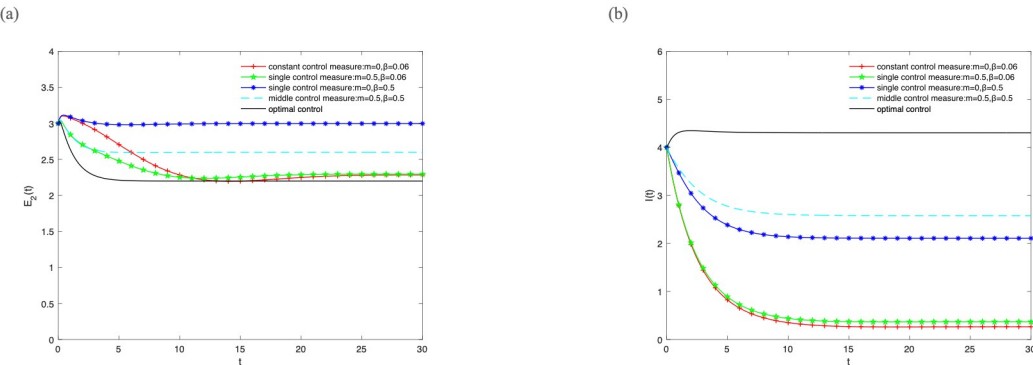

**Fig 6. The densities of (A) $E_2(t)$, (B) $I(t)$ change over time under different control strategies, where $\beta$ increases to 0.06 when the values of other parameters remain unchanged.**

out $R_0 = \rho(FV^{-1}) = \sqrt{\frac{B\alpha\beta(1+m)}{\mu(\beta+\gamma_1+\varepsilon+\mu)(\lambda+\mu)}}$, thereby calculating:

$$\frac{\partial R_0}{\partial m} = \frac{1}{2}\sqrt{\frac{\mu(\beta+\gamma_1+\varepsilon+\mu)(\lambda+\mu)}{B\alpha\beta(1+m)}} \times \frac{B\alpha\beta}{\mu(\beta+\gamma_1+\varepsilon+\mu)(\lambda+\mu)} > 0, \tag{58}$$

$$\frac{\partial R_0}{\partial \beta} = \frac{1}{2}\sqrt{\frac{\mu(\beta+\gamma_1+\varepsilon+\mu)(\lambda+\mu)}{B\alpha\beta(1+m)}}$$
$$\times \frac{B\alpha(1+m)(\mu\gamma_1\lambda+\mu^2\gamma_1+\mu\varepsilon\lambda+\mu^2\varepsilon+\mu^2\lambda+\mu^3)}{\mu(\beta+\gamma_1+\varepsilon+\mu)(\lambda+\mu)} > 0. \tag{59}$$

As can be seen, $R_0$ increases along with $m$. This indicates the authoritative "Super transmitters" existing in the social system can promote the transmission of information. In other words, the more authoritative "Super transmitters", the more expansive the transmitting range of information. Meanwhile, $R_0$ is also positively correlated with $\beta$. This demonstrates that when the hesitant individuals affected by the authoritative "Super transmitters" have a higher acceptance level of information, their probability of translating into information transmitters will be higher, and the transmission range of information will be wider.

The sensitivity analysis of $R_0$ is shown in Fig 7.

## 7 Conclusions

To investigate the effects of "Super transmission" and "Asymptomatic infection" on information transmission, this paper developed an *S2EIR* model that included the "Super transmitters" and "Asymptomatic carriers", which can reflect the transmitters' authoritativeness and individual acceptance level. After analyzing the equilibrium point and stability of the model and verifying the existence of optimal control, an optimal control strategy was proposed, and its effectiveness was validated through numerical simulation. Ultimately, the sensitivity of the optimal control parameters was analyzed.

Through the research in this paper, we have come to the following results:

1. The "Super transmission" and "Asymptomatic infection" phenomena that have been valued by the transmission of COVID-19, which still have a significant impact on information

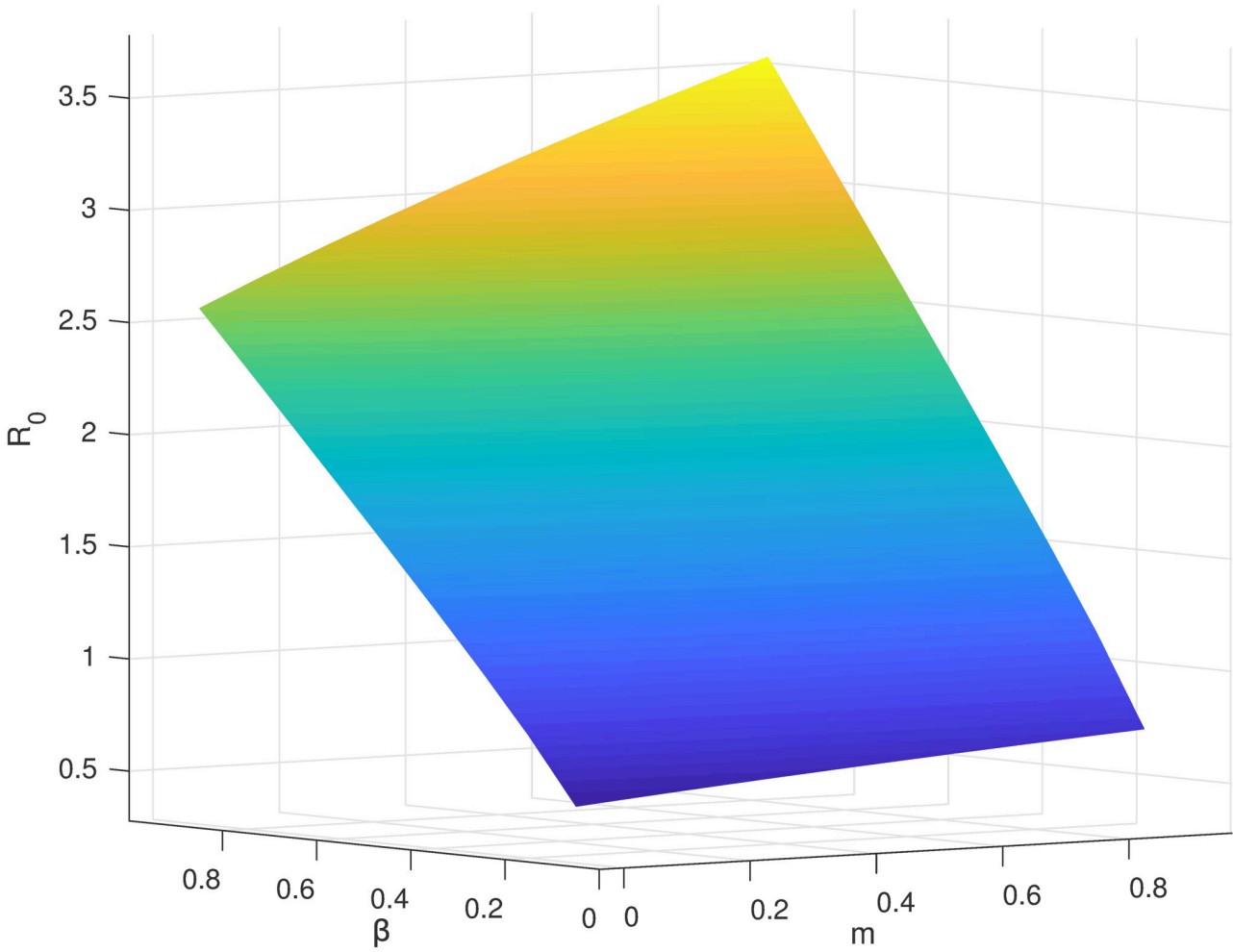

**Fig 7. The sensitivity analysis of the basic reproduction number $R_0$.**

transmission. We conduct a quantitative comparison of these two concepts in terms of information transmission. In contrast to traditional studies, our optimal control strategies are based on the optimal value of control variables;

2. The authoritative "Super transmitters" can promote the transmission of information and increase the acceptance level of information by hesitant individuals, making contact with those "Super transmitters" prone to becoming the transmitters of information. Thus, information transmission can be accelerated by increasing the proportion of "Super transmitters" among the population;

   Due to the unique nature of information, it is difficult for individuals who have not come into contact with "Super transmitters" to receive information through self-learning. Therefore, their acceptance level of information is generally low. In this case, these individuals may not be well-informed and thus are not active in information transmission;

3. The individual's self-learning and understanding ability guarantee the individuals to receive and transmit information. A greater proportion of hesitant individuals will emerge as information transmitters when their acceptance level of information is improved. To this end,

we can strengthen education or popularize knowledge to improve the understanding abilities of hesitant individuals.

The transmission of knowledge, technology, and skill-based information has contributed positively to social development. Expanding the transmission of such information and encouraging people to accept it will have a far-reaching impact on social progress. Our research discovered that information could be effectively disseminated by increasing the proportion of authoritative "Super transmitters" in the social system and improving individuals' learning abilities through education or knowledge popularization.

We will focus our future research on the following three aspects. First, we will consider the transmission of multi-information in the social system. Second, we will further study the influence of authoritative "Super transmitters" on transmitting various types of information over the same period. Finally, the interference mechanism of multi-information will be introduced to determine an approach for the stable transmission of information within the social system.

## Author Contributions

**Conceptualization:** Sida Kang, Xilin Hou.

**Data curation:** Sida Kang, Yuhan Hu.

**Formal analysis:** Sida Kang, Yuhan Hu.

**Investigation:** Sida Kang, Yuhan Hu.

**Methodology:** Sida Kang, Yuhan Hu.

**Software:** Sida Kang, Yuhan Hu.

**Validation:** Sida Kang, Yuhan Hu.

**Writing – original draft:** Sida Kang.

**Writing – review & editing:** Xilin Hou, Hongyu Liu.

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
