## [Decision Letter · Decision Letter 0]

14 Mar 2022

PONE-D-22-05396Dynamical analysis and optimal control for an information transmission model considering the phenomena of “Super Transmission” and “Asymptomatic Infection”PLOS ONE

Dear Dr. Hou,

Thank you for submitting your manuscript to PLOS ONE. After careful consideration, we feel that it has merit but does not fully meet PLOS ONE’s publication criteria as it currently stands. Therefore, we invite you to submit a revised version of the manuscript that addresses the points raised during the review process.

ACADEMIC EDITOR: The manuscript needs minor revision according to the reviewers' comments.

We look forward to receiving your revised manuscript.

Kind regards,

Mohammed S. Abdo

Academic Editor

PLOS ONE

Journal Requirements:

Additional Editor Comments:

Dear Authors,

we have received the reports from our advisors on your manuscript,

"Dynamical analysis and optimal control for an information transmission model considering the phenomena of "Super Transmission"and "Asymptomatic Infection", which you submitted to "PLOS ONE". Based on the advice received, I have decided that your manuscript could be reconsidered for publication should you be prepared to incorporate minor revisions. When preparing your revised manuscript, you are asked to carefully consider the reviewer's comments which can be found in the system and submit a list of responses to the comments and mark them in red if possible.

The manuscript needs to address the reviewers' comments.

Best regards

Mohammed S. Abdo

Academic Editor

Reviewers' comments:

Reviewer's Responses to Questions

**Comments to the Author**

1. Is the manuscript technically sound, and do the data support the conclusions?

Reviewer #1: Yes

Reviewer #2: Yes

2. Has the statistical analysis been performed appropriately and rigorously? 

Reviewer #1: Yes

Reviewer #2: N/A

3. Have the authors made all data underlying the findings in their manuscript fully available?

Reviewer #1: Yes

Reviewer #2: Yes

4. Is the manuscript presented in an intelligible fashion and written in standard English?

Reviewer #1: Yes

Reviewer #2: Yes

5. Review Comments to the Author

Reviewer #1: Review report

PLOS ONE

Manuscript Number: PONE-D-22-05396

Dynamical analysis and optimal control for an information transmission model

considering the phenomena of “Super Transmission” and “Asymptomatic

Infection”

by

Sida Kang, Xilin Hou, Yuhan Hu, Hongyu Liu

1 Summary and Recommendation:

The authors compared the phenomenon of "Super transmission" and "Asymp-

tomatic infection" in COVID-19 transmission to information transmission. The

former is similar to authoritative information transmission individuals, whereas

the latter is similar to individuals with low acceptance in information trans-

mission. Also, they constructed an S2EIR model with transmitter authority

and individual acceptance levels. Then, they analyzed the asymptotic stability

of information-free and information-existence equilibrium on a local and global

scale, as well as the model’s basic reproduction number, R0. Based on the Pon-

tryagin maximum principle, an optimal control strategy is designed to e¤ectively

facilitate information transmission. The numerical simulation corroborates the

theoretical analysis results and the system’s sensitivity to control parameter

changes. This work is well written and the results are new. Before the article

can be published, some points need to some minor revision:

1.1 Comments

Following misprints or suggestions are observed. The authors should study the

paper carefully for other possible typos.

1) Make sure all equations are properly cited

2) Introductions need to be improved with more details of current work.

Please mention which challenges you face to prove the results.

3) What is the di¤erence between stability and asymptotic stability?

4) It is very important to know the advantages of the present study? So, the

introduction needs to improve by highlighting the main advantages?.

5) What are the means of the matrixes F and V in equation 6. and how do

obtain them? please explain it.

6) How obtain equation 7? add some information.

7) In system 8, replace S;E1;E2; I;R with S;E1 ;E2 ; I;R:

8) How was equation 29 chosen in this form? which conditions followed for

it?

9) I suggest to improve the introduction section by studying some useful re-

cent papers/books, for example: https://doi.org/10.1016/j.chaos.2020.109867,

https://doi.org/10.1016/j.chaos.2021.110931, https://doi.org/10.1016/j.rinp.2021.104045.

‘10) Update references according to the style of journal.

Overall, the work is well written and the results are interesting.

1.2 My Recommendation

I would suggest that the paper should be accepted with minor revision due to

some of the corrections I pointed out above and in order to raise the standard

of this paper. The English need to be polished, punctuation mark needs to be

administered after each equation. Finally, I will be available for further revision

of this paper. After the authors take into account the suggestions as above I

recommend the publication of the paper.

2

Reviewer #2: In this paper, the authors compared the phenomenon of “Super transmission” and “Asymptomatic infection” in COVID-19 transmission to information transmission. They then constructed an S2EIR model with transmitter authority and individual acceptance levels and analyzed the asymptotic stability, as well as the model’s basic reproduction number.

Moreover, an optimal control strategy was designed based on the Pontryagin maximum principle. Finally, the numerical simulation is presented.

This paper is very interesting to read. The analysis in this paper is very good. The results are original and present a good degree of novelty. The techniques in this paper present are well-employed to obtain the intended results, and the proofs are correct.

This paper needs a minor revision, and I would like to recommend for accepting this paper after the following comments:

1) There are some typos and grammatical errors in some parts of this text, especially in the introduction section. Please double-check all sentences and correct all sentences that need to be corrected grammatically.

2) Please pay attention to all punctuation marks in the text.

3) Update the recent references related to this work; Chaos, Solitons & Fractals, 135, 109867.‏ https://doi.org/10.1016/j.chaos.2020.109867; Axioms 2021, 10(3), 228; https://doi.org/10.3390/axioms10030228.

4) I suggest that the authors amend the article title as follows:

"Dynamical analysis and optimal control of the developed information transmission model".

6. PLOS authors have the option to publish the peer review history of their article (what does this mean?). If published, this will include your full peer review and any attached files.

Reviewer #1: **Yes: **Mohammed A. Almalahi

Reviewer #2: No

---

## [Author Response · Author response to Decision Letter 0]

29 Mar 2022

Dear Editors and Reviewers:

 Thank you very much for your letter and the reviewers’ comments concerning our manuscript entitled "Dynamical analysis and optimal control for an information transmission model considering the phenomena of 'Super Transmission' and 'Asymptomatic Infection'" (ID: PONE-D-22-05396). These comments are indeed extremely helpful for revising and improving our paper, as well as the important guiding significance to our researches. We have studied these comments with care and have made correction which we hope meet with approval. The main corrections in the paper and the responds to the reviewer’s comments are as flowing:

 Responds to the reviewer’s comments:

Reviewer #1: 

1. Response to comment: (Make sure all equations are properly cited.)

Response: Thank you for helping us to find the errors and omissions. We are very sorry for not being careful to make this obvious mistake. According to your suggestions and requirements, we have corrected the mistakes cited in the original manuscript. At the same time, according to the whole comments, we have added some equations. Finally, we have checked all equations cited in the manuscript, and then make sure all equations are properly cited in our revised manuscript.

2. Response to comment: (Introductions need to be improved with more details of current work. Please mention which challenges you face to prove the results.)

Response: Thank you very much for your incisive comments and thorough reminders. I'm very sorry for the inaccurate and unclear expression due to our negligence. According to your comments and suggestions, we have improved with more details of current work and mentioned the challenges we face to prove the results in our revised manuscript(Page 3, Line 82-99). The details as following:

“The scholars mentioned above have conducted extensive research on transmitting various types of information via various channels, including the transmission of rumors[4], false information[5], competitive information[28] and malware[27]. Scholars who consider information transmission from the perspective of individuals are usually based on the attitude of transmission individuals to an event[22-24,31,32]. Few scholars consider the transmission individuals' attributes. Generally, most scholars primarily focus on controlling and transmitting negative information in networks[44,46]. On the other hand, confidential information must be transmitted via the contact for information that must be taught in person, such as high technology or practical skills. At the same time, due to the unique characteristics of this type of information, as well as the requirements for the transmitter's authority and the recipient's ability to accept it, there is a shortage of literature on the subject. In addition, the spread of COVID-19 is a hot topic in current research. However, few scholars consider the phenomenon of "Asymptomatic infection" in the transmission model. The research on the transmission of the phenomenon of "Asymptomatic infection" is even less, especially in information transmission. Therefore, consider the phenomenon of "Asymptomatic infection" in the information transmission model and investigate the impact of "Asymptomatic infection" on the information transmission system.”

3. Response to comment: (What is the difference between stability and asymptotic stability?)

Response: Thanks for your sincere comments and reminders, which are indeed to the point. Stability and asymptotic stability are important theories in the field of differential equations. It has a very important application in the study of the dynamic system in infectious disease transmission or information transmission. These theories are also applied in our paper. Therefore, we consult some materials to learn the difference between stability and asymptotic stability.

If for any given \\varepsilon>0 and t_0\\geq0 all exist \\delta=\\delta(\\varepsilon,t_0)>0, so that as long as x0 is satisfied:

x0-x1<δ ,

then we can get

x(t,t0,x0)-φ(t,t0,x1)<ε .

For every t\\geq t_0 establish, then the solution x=\\varphi(t,t_0,x_1) of differential equation \\frac{dx}{dt}=f(t,x) is said to be stability.

Suppose that x=\\varphi(t,t_0,x_1) is stable, and there is \\delta_1(0<\\delta_1<\\delta), so that as long as x0 satisfies:

x0-x1<δ1 ,

then we can get

limt→∞x(t,t0,x0)-φ(t,t0,x1)=0 .

Then the solution x=\\varphi(t,t_0,x_1) of differential equation \\frac{dx}{dt}=f(t,x) is said to be asymptotic stability.

It can be seen that asymptotic stability is a special case of stability. Asymptotic stability is more stringent than stability for the system. Stability requires that the state trajectory converge to a certain range near the equilibrium point. The asymptotic stability requires that the state trajectory converges to the equilibrium point on the premise of stability. The differential equation constructed in this paper has an equilibrium point, and the system finally converges to the equilibrium point. Therefore, the model in this paper meets the requirements of asymptotic stability. We think the model constructed in this paper needs the system to reach an asymptotically stability state. Asymptotic stability can better reflect the state of the model. Your comments have prompted us to deepen our understanding of the concepts of stability and asymptotic stability.

4. Response to comment: (It is very important to know the advantages of the present study? So, the introduction needs to improve by highlighting the main advantages.)

Response: Thanks for your sincere comments and reminders. Your suggestions are of great help to the improvement of the paper. According to your comments and suggestions, we have improved by highlighting the main advantages in our revised manuscript(Page 3, Line 100-Page 4, Line 119). The details as following:

“As a result of the above considerations, this paper compares the phenomena of "Super transmission" and "Asymptomatic infection" in COVID-19 transmission to information transmission. It then proposes an S2EIR model that incorporates the phenomena of "Super transmission" and "Asymptomatic infection". During the transmission of COVID-19, the phenomena of "Super transmission" and "Asymptomatic infection" are common. The term "Super transmission" refers to the phenomenon in which highly contagious individuals are more likely to spread the virus to the majority of patients. The term "Asymptomatic infection" refers to a situation in which a patient has been infected with a virus and has become a carrier of the virus but does not exhibit obvious disease symptoms due to individual resistance and physical quality differences. Both of these phenomena frequently occur during information transmission. "Super transmitters" are analogous to authoritative individuals in information transmission. The information transmission by such individuals is more easily accepted. Whereas "Asymptomatic infected individuals" are analogous to hesitant individuals with a low level of acceptance. However, this group of people has certain infectivity in the transmission of COVID-19. However, individuals who conceal information rarely choose to share it with others during the transmission of information. At the same time, others are unaware that such individuals know the information. This is also the manifestation of "Asymptomatic" in information transmission. The optimal control strategy of information transmission is quantified by scientific methods to test these phenomena.”

5. Response to comment: (What are the means of the matrixes F and V in equation 6, and how do obtain them? please explain it.)

Response: Thank you very much for your comments! In the original manuscript, perhaps our expression is not comprehensive and careful enough. We would like to make a further explanation on this point.

 It is explained in the reference (https://doi.org/10.1016/j.rinp.2021.104045) you recommend, F and V represent the infection and transition matrices respectively. Meanwhile, we have modified and added the method of obtaining the matrixed F and V in the revised manuscript (Page 6, Line 190-193). The details as following:

“Calculate the Jacobian matrices of F(X) and V(X) in system (5) respectively, and then take the sub matrices corresponding to the first two variables (i.e. E1, I) directly related to the number of communicators. The results are as follows:

F=\\left(\\begin{matrix}0&\\alpha(1+m)S\\\\\\beta&0\\\\\\end{matrix}\\right)\\ ,\\ V=\\left(\\begin{matrix}\\beta+\\gamma_1+\\varepsilon+\\mu&0\\\\0&\\lambda+\\mu\\\\\\end{matrix}\\right)\\ , (6)

 where F and V represent the infection and transition matrices respectively[38].”

6. Response to comment: (How obtain equation 7? add some information.)

Response: Thank you very much for the attentive and earnest comments! We do regret for the lack of smooth logical narration due to the omission of necessary steps and instructions in the original manuscript, which has brought you a dissatisfactory reading experience.

We have modified and added relevant parts in the revised manuscript (Page 6, Line 193-198). The details as following:

(Tip: Due to the addition of equation, equation 7 in the original manuscript has changed to equation 9 in the revised manuscript.)

“By simple calculation, the inverse matrix of V can be obtained as:

V^{-1}=\\left(\\begin{matrix}\\frac{1}{\\beta+\\gamma_1+\\varepsilon+\\mu}&o\\\\o&\\frac{1}{\\lambda+\\mu}\\\\\\end{matrix}\\right)\\ . (7)

The next generation matrix[47] is 

{FV}^{-1}=\\left(\\begin{matrix}0&\\frac{\\alpha(1+m)S}{\\lambda+\\mu}\\\\\\frac{\\beta}{\\beta+\\gamma_1+\\varepsilon+\\mu}&0\\\\\\end{matrix}\\right)\\ . (8)

Hence, the basic reproduction number R0 of system (1) is the spectral radius of the next generation matrix FV-1. Here, the spectral radius is the maximum value of characteristic root of FV-1. Therefore, R0 can be computed as:

R_0=\\rho({FV}^{-1})=\\sqrt{\\frac{B\\alpha\\beta(1+m)}{\\mu(\\beta+\\gamma_1+\\varepsilon+\\mu)(\\lambda+\\mu)}}\\ . (9)’’

7. Response to comment: (In system 8, replace S; E1; E2; I; R with S*; E_1^\\ast; E_2^\\ast; I*; R*.)

Response: Thank you for helping us to find the omissions. We are very sorry for not being careful to make this obvious mistake. According to your suggestions and requirements, we have replaced S; E1; E2; I; R with S*; \\mathbit{E}_\\mathbf{1}^\\ast; \\mathbit{E}_\\mathbf{2}^\\ast; I*; R* in system 8 in our revised manuscript (Page 6, Line 203).

(Tip: Due to the addition of equation, system 8 in the original manuscript has changed to system 10 in the revised manuscript.)

8. Response to comment: (How was equation 29 chosen in this form? which conditions followed for it?)

Response: Thanks for your sincere comments and reminders, which are indeed to the point. According to our understanding，there is no unified standard and rule for constructing Lyapunov function. According to the conditions to be satisfied by Lyapunov function, we have made a lot of attempts. The method we choose just satisfy the conditions of Lyapunov function. Therefore we construct the Lyapunov function as: 

(Tip: equation 29 has been changed to equation 31 due to the change of equation number. We apologize for the inconvenience.).

Wt=[St-S*+E1t-E1*+E2t-E2*+It-I*⬚+Rt-R*]2, (31)

and

\\ \\ \\ \\ \\ \\ \\ \\ \\ \\ \\ \\ \\ \\ \\ W\\prime\\left(t\\right)=2\\left[\\left(S\\left(t\\right)-S^\\ast\\right)+\\left(E_1\\left(t\\right)-E_1^\\ast\\right)+\\left(E_2\\left(t\\right)-E_2^\\ast\\right)+\\left(I\\left(t\\right)-I^\\ast\\right)\\ \\ \\ \\ \\ \\ \\ \\ \\ +\\left(R\\left(t\\right)-R^\\ast\\right)\\right]\\left[S\\prime\\left(t\\right)+E_1\\prime\\left(t\\right)+E_2\\prime\\left(t\\right)+I\\prime\\left(t\\right)+R\\prime\\left(t\\right)\\right]=2\\left[\\left(S\\left(t\\right)-S^\\ast\\right)+\\left(E_1\\left(t\\right)-E_1^\\ast\\right)+\\left(E_2\\left(t\\right)-E_2^\\ast\\right)+\\left(I\\left(t\\right)-I^\\ast\\right)+\\left(R\\left(t\\right)-R^\\ast\\right)\\right]\\left[B-\\mu S-\\mu E_1-\\mu E_2-\\mu I-\\mu R\\right]\\ .\\ \\ \\ \\ \\ \\ \\ \\ \\ \\ \\ \\ \\ \\ \\ \\ \\ \\ \\ \\ \\ \\ \\ \\ \\ \\ \\ (32)

Because of the existence of E^\\ast\\left(S^\\ast,E_1^\\ast,E_2^\\ast,I^\\ast,R^\\ast\\right), we can know that B-\\mu S^\\ast-\\mu{E_1}^\\ast-\\mu{E_2}^\\ast-\\mu I^\\ast-\\mu R^\\ast=0, so that B=\\mu S^\\ast+\\mu{E_1}^\\ast+\\mu{E_2}^\\ast+\\mu I^\\ast+\\mu R^\\ast.

Then, Eq.(32) can be written as:

W\\prime\\left(t\\right)=2\\left[\\left(S\\left(t\\right)-S^\\ast\\right)+\\left(E_1\\left(t\\right)-E_1^\\ast\\right)+\\left(E_2\\left(t\\right)-E_2^\\ast\\right)+\\left(I\\left(t\\right)-I^\\ast\\right)\\ \\ \\ \\ \\ \\ \\ \\ \\ +\\left(R\\left(t\\right)-R^\\ast\\right)\\right]\\left[\\mu S^\\ast+\\mu{E_1}^\\ast+\\mu{E_2}^\\ast+\\mu I^\\ast+\\mu R^\\ast-\\mu S-\\mu E_1-\\mu E_2-\\mu I-\\mu R\\right]=-2\\left[\\left(S-S^\\ast\\right)+\\left(E_1-E_1^\\ast\\right)+\\left(E_2-E_2^\\ast\\right)+\\left(I-I^\\ast\\right)+\\left(R-R^\\ast\\right)\\right]^2\\le0\\ .\\ \\ \\ \\ \\ \\ \\ \\ \\ \\ \\ \\ \\ \\ \\ \\ \\ \\ \\ \\ \\ \\ \\ \\ \\ \\ \\ \\ \\ \\ \\ \\ \\ \\ \\ \\ \\ \\ \\ \\ \\ \\ \\ \\ \\ \\ \\ \\ \\ \\ \\ \\ \\ \\ \\ \\ \\ \\ \\ \\ \\ \\ \\ \\ \\ \\ \\ \\ \\ \\ \\ \\ \\ \\ \\ \\ \\ \\ \\ \\ \\ \\ \\ \\ \\ \\ \\ \\ \\ \\ \\ \\ \\ \\ \\ \\ \\ \\ \\ \\ \\ \\ \\ \\ \\ \\ (33)

Besides that, W\\prime\\left(t\\right)=0 holds if and only if S\\left(t\\right)=S^\\ast,E_1\\left(t\\right)=E_1^\\ast,E_2\\left(t\\right)=E_2^\\ast,I\\left(t\\right)=I^\\ast,R\\left(t\\right)=R^\\ast. Hence, the information-existence equilibrium point E^\\ast\\left(S^\\ast,E_1^\\ast,E_2^\\ast,I^\\ast,R^\\ast\\right) of system (1) is globally asymptotically stable based on Lyapunov-LaSalle Invariance Principle[48].

We choose a simple and effective way to construct Lyapunov function in this paper. We will try our best to study the construction of Lyapunov function by other methods in the future.

9. Response to comment: (I suggest to improve the introduction section by studying some useful recent papers/books, for example: https://doi.org/10.1016/j.chaos.2020.109867, https://doi.org/10.1016/j.chaos.2021.110931, https://doi.org/10.1016/j.rinp.2021.104045.)

Response: Thanks for your comments and suggestions. We have downloaded and read these references. These references are closely related to the research of this paper, and they deepen our understanding. It is more helpful for us to clearly understand the characteristics of virus transmission and the choice of methods. Therefore, we have added refers to these useful recent papers in the revised manuscript (Page 2, Line 56-Page 3, Line 67). The details are as follows:

“Abdo et al. analyzed and found the solution for the model of nonlinear fractional differential equations describing the deadly and most parlous virus, so-called coronavirus (COVID-19). The study discovered that the susceptibility decreases more rapidly at the lower fractional order of the derivative. Similarly, the increase in infections is also rapid, but in a smaller order[37]. Almalahi et al. used fractal-ABC type fractional differential equations by incorporating population self-protection behavior changes to study the dynamics of 2019-nCoV transmission[38], and investigated sufficient conditions of existence and uniqueness of positive solutions for a finite system of \\varphi-Hilfer fractional differential equations[39]. Jeelani et al. investigated a fractional-order mathematical model of COVID-19[40]. These research findings are critical in the prediction of 2019-nCoV.”

10. Response to comment: (Update references according to the style of journal.)

Response: Thanks very much for your comments. According to your request, we have checked upon and updated references according to the style of journal. And we have already put them right in the revised version. As these minor errors are more, we have not marked each and every of them corresponding page and line numbers in details.

 Special thanks to you for your good comments!

Reviewer #2: 

1. Response to comment: (There are some typos and grammatical errors in some parts of this text, especially in the introduction section. Please double-check all sentences and correct all sentences that need to be corrected grammatically.)

Response: Thanks very much for your comments. According to your request, we have checked upon a number of grammatical and typo mistakes once again. And we have already put them right in the revised version. As these minor errors are more, we have not marked each and every of them corresponding page and line numbers in details.

2. Response to comment: (Please pay attention to all punctuation marks in the text.)

Response: Thanks very much for your comments. Your comments are conducive us to be more careful. It is more helpful for us to improve our good scientific research literacy. According to your request, we have checked upon all punctuation once again. And we have already put them right in the revised version. As these minor errors are more, we have not marked each and every of them corresponding page and line numbers in details.

3. Response to comment: (Update the recent references related to this work; Chaos, Solitons & Fractals, 135, 109867.‏ https://doi.org/10.1016/j.chaos.2020.109867; Axioms 2021, 10(3), 228; https://doi.org/10.3390/axioms10030228.)

Response: Thanks for your comments and suggestions. We have downloaded and read these references. These references are closely related to the research of this paper, and they deepen our understanding. It is more helpful for us to clearly understand the characteristics of virus transmission and the choice of methods. Therefore, we have added refers to these useful recent references in the revised manuscript (Page 2, Line 56-Page 3, Line 67). The details are as follows:

“Abdo et al. analyzed and found the solution for the model of nonlinear fractional differential equations describing the deadly and most parlous virus, so-called coronavirus (COVID-19). The study discovered that the susceptibility decreases more rapidly at the lower fractional order of the derivative. Similarly, the increase in infections is also rapid, but in a smaller order[37].

Jeelani et al. investigated a fractional-order mathematical model of COVID-19[40]. These research findings are critical in the prediction of 2019-nCoV.”

4. Response to comment: (I suggest that the authors amend the article title as follows:

"Dynamical analysis and optimal control of the developed information transmission model".)

Response: Thanks for your sincere comments and reminders. Compared with our original title, the title you proposed is more concise and more widely applicable. According to your suggestions and requirements, we have amended the article title as "Dynamical analysis and optimal control of the developed information transmission model" in our revised manuscript.

 Special thanks to you for your good comments!

 We tried our best to improve the manuscript and made some changes in the manuscript. These changes will not influence the content and framework of the paper. 

 We appreciate for Editors/Reviewers’ warm work earnestly, and hope that the correction will meet with approval.

 Once again, thank you very much for your comments and suggestions.

---

## [Decision Letter · Decision Letter 1]

27 Apr 2022

Dynamical analysis and optimal control of the developed information transmission model

PONE-D-22-05396R1

Dear Dr. Hou,

We’re pleased to inform you that your manuscript has been judged scientifically suitable for publication and will be formally accepted for publication once it meets all outstanding technical requirements.

Kind regards,

Mohammed S. Abdo

Academic Editor

PLOS ONE

Additional Editor Comments (optional):

The authors did their best to improve the manuscript and made some changes to the manuscript according to the reviewers' reports. These changes did not affect the content and frame of the paper.

Therefore, **I decided to accept the manuscript provided that the authors delete some of the references suggested by the reviewers and keep the references related to the topic of the paper.**

Reviewers' comments:

Reviewer's Responses to Questions

**Comments to the Author**

1. If the authors have adequately addressed your comments raised in a previous round of review and you feel that this manuscript is now acceptable for publication, you may indicate that here to bypass the “Comments to the Author” section, enter your conflict of interest statement in the “Confidential to Editor” section, and submit your "Accept" recommendation.

Reviewer #1: All comments have been addressed

Reviewer #2: All comments have been addressed

2. Is the manuscript technically sound, and do the data support the conclusions?

Reviewer #1: Yes

Reviewer #2: Yes

3. Has the statistical analysis been performed appropriately and rigorously? 

Reviewer #1: Yes

Reviewer #2: N/A

4. Have the authors made all data underlying the findings in their manuscript fully available?

Reviewer #1: Yes

Reviewer #2: Yes

5. Is the manuscript presented in an intelligible fashion and written in standard English?

Reviewer #1: Yes

Reviewer #2: Yes

6. Review Comments to the Author

Reviewer #1: The authors compared the phenomenon of "Super transmission" and "Asymp-

tomatic infection" in COVID-19 transmission to information transmission. The

former is similar to authoritative information transmission individuals, whereas

the latter is similar to individuals with low acceptance in information trans-

mission. Also, they constructed an S2EIR model with transmitter authority

and individual acceptance levels. Then, they analyzed the asymptotic stability

of information-free and information-existence equilibrium on a local and global

scale, as well as the model’s basic reproduction number, R0. Based on the Pon-

tryagin maximum principle, an optimal control strategy is designed to e¤ectively

facilitate information transmission. The numerical simulation corroborates the

theoretical analysis results and the system’s sensitivity to control parameter

changes. This work is well written and the results are new

The authors take into account the suggestions. All comments have been addressed. I recommend the publication of the paper.

Reviewer #2: Dear Authors, I carefully reviewed the revised manuscript PONE-D-22-05396R1 entitled “Dynamical analysis and optimal control of the developed information transmission model” and I found the following:

This paper is very interesting to read. The analysis in this paper is very good. The results are original and present a good degree of novelty. The techniques in this paper present are well-employed to obtain the intended results, and the proofs are correct.

So, the revised manuscript is suitable for publication.

7. PLOS authors have the option to publish the peer review history of their article (what does this mean?). If published, this will include your full peer review and any attached files.

Reviewer #1: No

Reviewer #2: No

---

## [Editor Report · Acceptance letter]

12 May 2022

PONE-D-22-05396R1 

Dynamical analysis and optimal control of the developed information transmission model 

Dear Dr. Hou:

I'm pleased to inform you that your manuscript has been deemed suitable for publication in PLOS ONE. Congratulations! Your manuscript is now with our production department. 

Kind regards, 

on behalf of

Dr. Mohammed S. Abdo 

Academic Editor

PLOS ONE